# New Mathblocks-Based Feistel-Like Ciphers for Creating Clone-Resistant FPGA Devices

**Saleh Mulhem *** and **Wael Adi**

Institute of Computer and Network Engineering, Technical University of Braunschweig,
38106 Braunschweig, Germany; w.adi@tu-bs.de
*   Correspondence: s.mulhem@tu-bs.de

**Abstract:** The Secret Unknown Cipher (SUC) concept was introduced a decade ago as a promising technique for creating pure digital clone-resistant electronic units as alternatives to the traditional non-consistent Physical Unclonable Functions (PUFs). In this work, a very special unconventional cipher design is presented. The design uses hard-core FPGA (Field Programmable Gate Arrays) -Mathblocks available in modern system-on-chip (SoC) FPGAs. Such Mathblocks are often not completely used in many FPGA applications; therefore, it seems wise to make use of such dead (unused) modules to fabricate usable physical security functions for free. Standard cipher designs usually avoid deploying multipliers in the cipher mapping functions due to their high complexity. The main target of this work is to design large cipher classes (e.g., cipher class size $>2^{600}$) by mainly deploying the FPGA specific mathematical cores. The proposed cipher designs are novel hardware-oriented and new in the public literature, using fully new unusual mapping functions. If a random unknown selection of one cipher out of $2^{600}$ ciphers is self-configured in a device, then a Secret Unknown Cipher module is created within a device, making it physically hard to clone. We consider the cipher module for free (for zero cost) if the major elements in the cipher module are making use of unused reanimated Mathblocks. Such ciphers are usable in many future mass products for protecting vehicular units against cloning and modeling attacks. The required self-reconfigurable devices for that concept are not available now; however, they are expected to emerge in the near future.

**Keywords:** Feistel-like cipher; secret unknown cipher; physical unclonable function; Latin square; involution; golden S-Boxes; FPGA system-on-chip

## 1. Introduction

One of the most significant security threats to emerging electronic devices is cloning or theft of identity. Therefore, the security requirements are steadily growing to face such threats and challenges. In the last decades, several proposals were introduced for identifying an electronic device by using a secret stored key in an embedded non-volatile memory (NVM) [1]. Unfortunately, such technology has been proven inefficient against physical attacks [2]. Alternatively, Physically Unclonable Functions (PUFs) were proposed to serve as unclonable identities for electronic devices [3,4] as an alternative to key storage in NVM [4]. The main idea of PUFs is to seek a physical mapping out of the intrinsic properties or physical structures of a device. However, major results showed that PUF technologies suffer from lacking consistency over a long period of time due to several factors such as noise, aging, metastability, sensitivity to temperature, supply voltage variations, and other factors [4]. Moreover, PUF output response bits suffer from being non-uniformly distributed [4], which leads to offering more correlations between PUF input/ output pairs or the so-called Challenge–Response Pairs (CPRs), in other words simplifying the modeling attacks. In this case, a set of CRPs can be given as a training set to a Machine learning (ML) algorithm, which constructs a predictive model of the PUF. For instance,

several predictive models of various PUFs were given with error rates less than 1% for Arbiter PUFs, 4.5 % for Feed Forward Arbiter PUFs, and less than 1% for Ring Oscillator PUFs [5].

The inconsistency behavior of PUF CRPs were counteracted by adding a complex error-correcting fuzzy extractor [6] using a helper data algorithm (HDA) [7] together with an error correction code (ECC) [4]. Therefore, most of the proposed PUFs are still very costly to implement.

Furthermore, PUFs can be perceived as a physically unclonable source of randomness. The random physical factors are initiated during manufacturing. These random and unpredictable factors provide a PUF structure with a high level of the obscurity, unknowingness, and making the PUF substantially impossible to clone [8]. Further, PUFs were gradually defined as a physical one-way function (POWF) [9], then as a controlled physical random function [10], and recently as a physical unknown function [11].

This paper introduces a new proposal of an unknown cipher/ function serving as a digital PUF. As the concept of secret unknown ciphers is not well known in the public literature, the paper's technical presentation is sequenced and organized as follows:

- The state of the art of currently used unknown functions such as PUFs, PUF-based unknown key generation, etc. is critically presented showing their vulnerabilities and drawbacks in Section 2.
- The concept of the Secret Unknown Ciphers (SUCs) and their creation process toward building clone-resistant devices are presented in Section 3.
- The usage of hard-core arithmetic modules in designing the proposed unknown ciphers/functions are explained in Section 4 based on the existing resources of a modern FPGA technology such as Microsemi Smart-Fusion®2.
- New cipher classes based on mainly deploying multipliers in the ring of integers as Feistel-like ciphers classes are presented in Section 5.
- Sample hardware modeling and a complexity evaluation of such cipher classes and their security analysis are discussed and investigated in Sections 6–8.

**New Contributions:** This work is a new approach and improvement to follow our recent publications [12–14] toward developing unknown cipher functions to serve as digital PUFs. This paper introduces novel Mathblocks-based involutions for a Feistel-like class of new Secret Unknown Ciphers (SUCs) structures within modern FPGA devices. The new involutions are proved to exhibit efficient combined confusion and diffusion at the same time. The overall resulting security quality and implementation efficiency is shown to exceed the conventional cipher structures in anti-cloning applications.

## 2. State of the Art on Unclonable Electronic Units

In the following, PUFs and other proposals based on PUFs are summarized as unknown physical functions. Firstly, intrinsic PUFs [3] as unknown functions are presented. Then, PUF-based unknown key generation for a block cipher, especially for a pseudo-random function (PRF) [15], are investigated. Finally, a block cipher deploying PUFs [16] as an unknown function is presented. The following three technical discussions on PUFs are presented with some details as being closely related to the objectives of our cipher proposal.

### 2.1. PUFs as Unkown/Random Functions

This is a case of describing PUFs as unknown functions, where a formalization of a PUF technology can be presented and described as follows:

**Definition 1.** *Let $\Gamma_{PUF}$ be a set of certain PUFs, and $PUF \in \Gamma_{PUF}$. Then, PUF is defined as a mapping that is easy to compute, hard to invert, unpredictable, and derived from the random behavior of a complex physical object/device:*

$$PUF : \{0, 1\}^* \to \{0, 1\}^* \tag{1}$$

*where \* is a Kleene star and $\{0,1\}^*$ is the set of all possible binary strings with finite lengths. Note that a PUF is fundamentally considered to be a mapping from $\{0,1\}^*$ to $\{0,1\}^*$ [9]. According to [17], if a PUF can respond to every challenge from $\{0,1\}^*$ by a response from $\{0,1\}^*$, then a PUF is a one-way function. However, PUFs practically do not fulfill the requirements of a one-way function [18]. In this case, a PUF is defined as a mapping from a finite domain to a finite range,*

$$PUF : \{0,1\}^m \rightarrow \{0,1\}^n. \tag{2}$$

Furthermore, the number of all possible PUFs is theoretically upper bounded by the number of all possible mappings from $\{0,1\}^m$ to $\{0,1\}^n$,

$$|\Gamma_{PUF}| \leq 2^{n \cdot 2^m}. \tag{3}$$

If $|\Gamma_{PUF}| = 2^{n \cdot 2^m}$, then PUFs are seen as unknown/random functions [19]. Unfortunately, there is no guarantee that a physical object/device can produce this huge number of distinct mappings. This can be deduced through the following approaches. First, it is assumed that a PUF is "an isolated physical system $S$ which fits into a sphere of radius $R$ [18]". The maximum entropy $H_S$ (information content) of PUFs is upper bounded in its volume, as follows [20],

$$H_S \leq \alpha \cdot R^2 \tag{4}$$

where the constant $\alpha$ is related to physical quantities such as the speed of light, etc. [18].

The second approach uses the information capacity concept [19]. In this case, PUF is defined as a silicon device that implements a deterministic function, where the silicon object consists of $N$ silicon cells, such as memory bits, flip-flops, etc. In this case, the maximum information capacity/entropy $I_{max}$ of such a PUF is given in [19] as,

$$I_{max} \leq N \cdot C \text{ bits} \tag{5}$$

where $C \ll 1$ is the information capacity of one cell.

According to Equations (4) and (5), it is concluded that the information capacity of PUFs is limited and upper bounded. Therefore, the cardinality of $\Gamma_{PUF}$ is upper bounded as,

$$|\Gamma_{PUF}| \ll 2^{n 2^m}. \tag{6}$$

This implies that a PUF from a $\Gamma_{PUF}$ is not a random function [19].

### 2.2. PUF-Based Unkown Key Generation for Pseudo-Random Fuctions

This is a case of using an unknown key for a known cipher [15] where a PUF-based key generation for a standard block cipher can be simply constructed by choosing a conventional cipher and taking the key source as the PUF output/response $K$ to some known input $Z_0$.

The resulting cipher with the unknown key $K$ behaves precisely as a randomly chosen function from $\{0,1\}^n$ to $\{0,1\}^m$. Therefore, the resulting cipher's behavior fulfills the requirements to represent a PRF. As the cipher is public, the attack complexity is $2^m$, where $m$ is the size of the unknown key $K$.

In [21], PUF-based key generation for cryptographic application was practically investigated. Sadeghi et al. [15] combined key storage and strong PUF to produce an unknown key for a cipher, as shown in Figure 1. This proposal achieves a high level of security, where the PUF's response approaches a PRF behavior, and it becomes hard to be impersonated or to be modeled. Unfortunately, the resulting cipher structure still requires an additional complex fuzzy extractor and helper data to make the PUF respond with a consistent output (unknown key $K$).

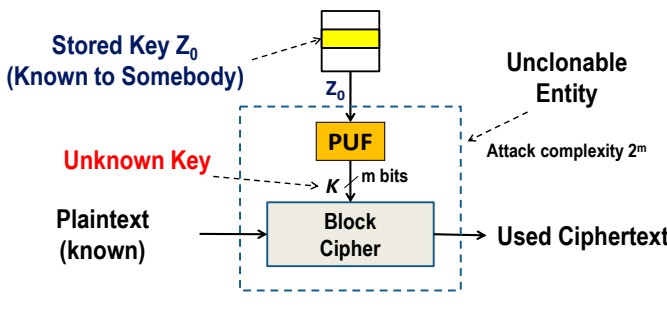

**Figure 1.** Randomizing Physical Unclonable Function (PUF) response by using the PUF as an unknown key source.

### 2.3. A Block Cipher Deploying PUFs as Unkown Round Functions

In this case, an unknown cipher is created by using PUFs as a part of the cipher mappings. In [16], a block cipher deploying PUFs was proposed, where a cipher is constructed as three cascaded Feistel cipher rounds with PUFs as round functions (see Figure 2). The resulting cipher fulfills the requirements of being a PRF.

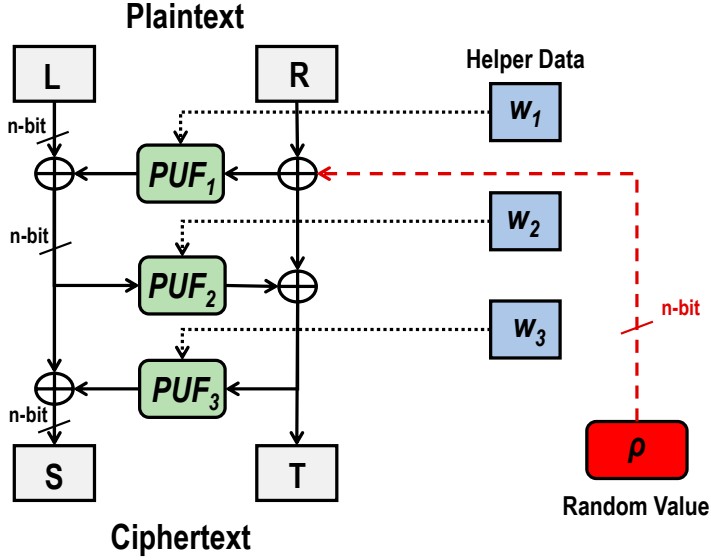

**Figure 2.** A randomized three-round Feistel cipher-based on PUFs [16].

In [19], Wu and O'Neill determined the necessary condition for a block cipher deploying PUFs to become a PRF. The results showed that such a cipher should have a high level of diffusion and confusion to become a PRF. In Figure 2, a Feistel cipher based on PUFs represents an efficient structure as a PRF, but it still requires additional helper data in addition to the PUFs to produce consistent outputs.

## 3. The Concept of Secret Unknown Ciphers Modules as PUF Alternatives

The unknown cipher concept is an entirely new security paradigm in the public literature. The unknown cipher here does not deal with protecting the communications or the links between at least two parties, as a sender and a receiver, which requires the cipher to be commonly known to both parties (Kerckhoffs's principle). In particular, the SUC is fundamentally designed for the identification process to serve as a clone-resistant identity [22]. We postulate that "unclonability" is only possible

if unknown structures are created. Therefore, a cipher designed to be embedded as a structure that is unknown to anybody (including its designer) does not violate Kerckhoff's principle. On the other hand, SUC should not be confused with "security by obscurity", where the cipher is designed by a cryptographer, known to the manufacturer, and then kept secret and obscure.

SUC creation is a very challenging task. Figure 3 illustrates the SUC creation concept in a non-volatile (NV) FPGA device having internal self-reconfiguration capability. A large class of ciphers $\{C_1, C_2 \ldots C_\sigma\}$ are first created $\sigma \to \infty$ and offered for selection. Then, a single-event process triggers a true random number generator (TRNG), leading to select randomly an unknown cipher choice $C_j$ from the infinite number $\sigma$ of the created distinct ciphers. After this process, all the dashed entities in Figure 3 are then irreversibly killed and fully removed from the chip.

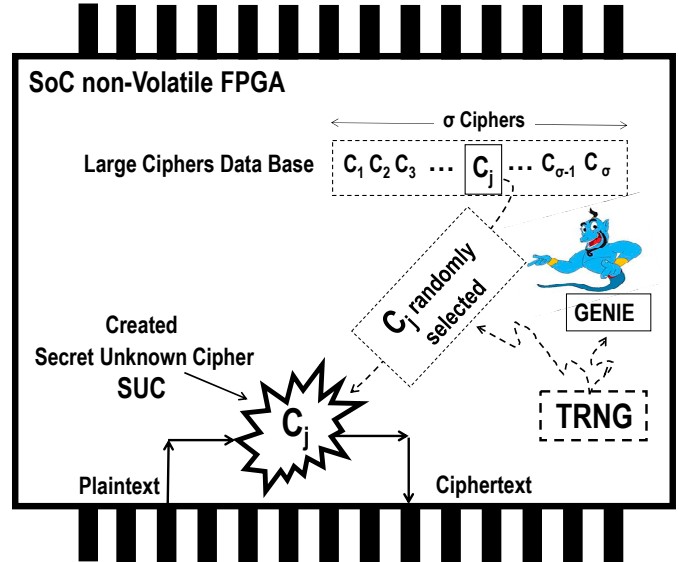

**Figure 3.** Key idea for generating a Secret Unknown Cipher (SUC).

The resulting cipher is a secret yet unknown cipher, and is a non-repeatable selection. It is even an unknown choice to the cipher designer/creator himself. The "Secret Unknown Cipher" (SUC) is realizable in an emerging VLSI device that allows self-creation of permanent unknown usable secret structures as "an electronic mutation", as indicated in [23]. Note that for the functionality of the concept, there is no need to publish the SUC creation procedure/program of the cipher class, which is designated from now on as the "GENIE" as a smart cipher designer. However, for worst-case security analysis, we assume that the cipher creating "GENIE" is published.

Following other cryptographers who use the term Oracle (inspired by the gods) to describe a theoretical black box model, the term GENIE is inspired from the oriental folk tales of *One Thousand and One Nights*. In the tales, a powerful ghost called GENIE can make all wishes come true; however, nobody knows how the GENIE can grant all wishes. In the ultimate case, our GENIE is a powerful cryptographer who can virtually create all possible ciphers of a given size.

### 3.1. Creation Concept of Unknown Ciphers as Clone-Resistant Entities/Modules

The proposed SUC is conceptually based on the following principle: "the only secret which can be kept unrevealed is the one which nobody knows" [13]. From a practical point of view, if the cipher creator itself cannot predict and foretell exactly the created cipher, then the cipher is considered as not known when the cipher class size $\sigma \to \infty$.

Figure 4 illustrates a possible SUC creation that is assumed to be processed in a secure environment. The process may proceed as follows:

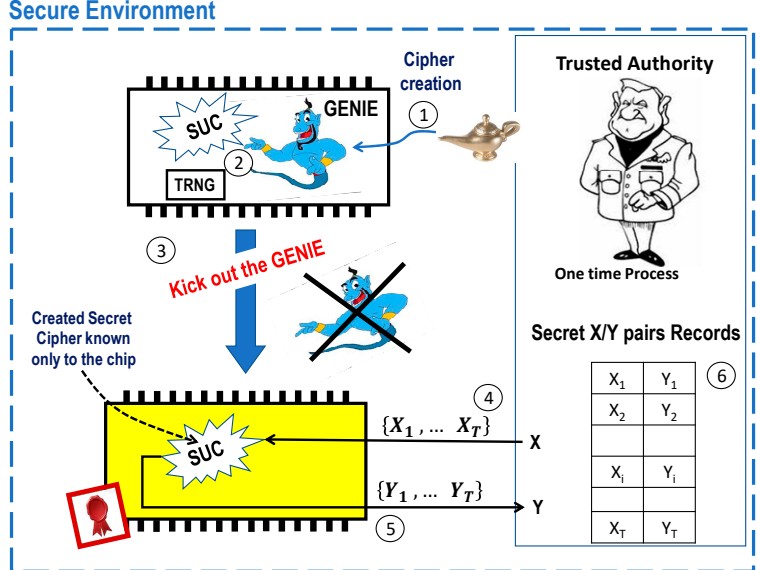

**Figure 4.** Mutating a Secret Unknown Cipher (SUC) into a system-on-chip (SoC) device.

**SUC creation phase:**

1. A trusted authority (TA) injects one-time into a system-on-chip (SoC) device the software package "GENIE" as an SUC creator for a short time (*as much time as required to create an unknown cipher, usually a few milliseconds*).
2. Then, the GENIE is internally triggered to generate/select a permanent and unpredictable secure cipher with the help of an internal, non-repeatable, unpredictable, and unknown bit stream from the in-chip TRNG.
3. After creating an SUC, the GENIE is completely and irreversibly deleted. What remains is a non-removable and unchangeable operational cipher (a SUC) that nobody knows.

**SUC personalization phase:**

4. TA randomly selects a set $\{x_1, \ldots x_T\}$ of cleartext vectors out of the $2^n$ possible combinations, where $n$ is the size of the SUC input/output space in bits.
5. TA stimulates the SoC device to encipher the cleartext vectors into the ciphertexts $\{y_1, \ldots y_T\}$ using its SUC within the device.
6. The resulting $T$-$(x_i, y_i)$ pairs are stored as secret pairs in the individual (personal) device records by the TA. The records have to be kept secret for later use.

As the created TRNG bits are fully and exclusively responsibly for creating the SUC, and as TRNG bits are unpredictable, non-repeatable, and unknown, the resulting created SUC in the SoC device is also unknown and unpredictable, such that:

$$SUC_t = GENIE(TRNG_t). \tag{7}$$

For every $t > 0$. This implies that

$$SUC_t : \{0,1\}^n \times \{0,1\}^{k_t} \to \{0,1\}^n \tag{8}$$

where $n$ is the bit size of the SUC input/output space and $k_t$ is the bit size of the cipher's secret key. In addition, SUC has the property of being able to generate a large number of distinct CRPs as cleartext/ciphertext pairs, which is upper bounded by $2^n$. This counteracts the lack of CR space in the case of traditional analog PUFs.

The created cipher $SUC_t$ is a result of the $TRNG_t$ random sequence that is not known to anybody. Moreover, it is highly probable that for any two-time points $t_1$ and $t_2$,

$$TRNG_{t_1} \neq TRNG_{t_2} \rightarrow SUC_{t_1} \neq SUC_{t_2}. \tag{9}$$

Therefore, each resulting SoC device has its individual SUC with a probability $\left(1 - \frac{1}{\sigma}\right) \rightarrow 1$.

**How to Use an SUC?**

Figure 5 shows a generic two-way identification protocol using such SUCs for authenticating a personalized $SoC_A$ device.

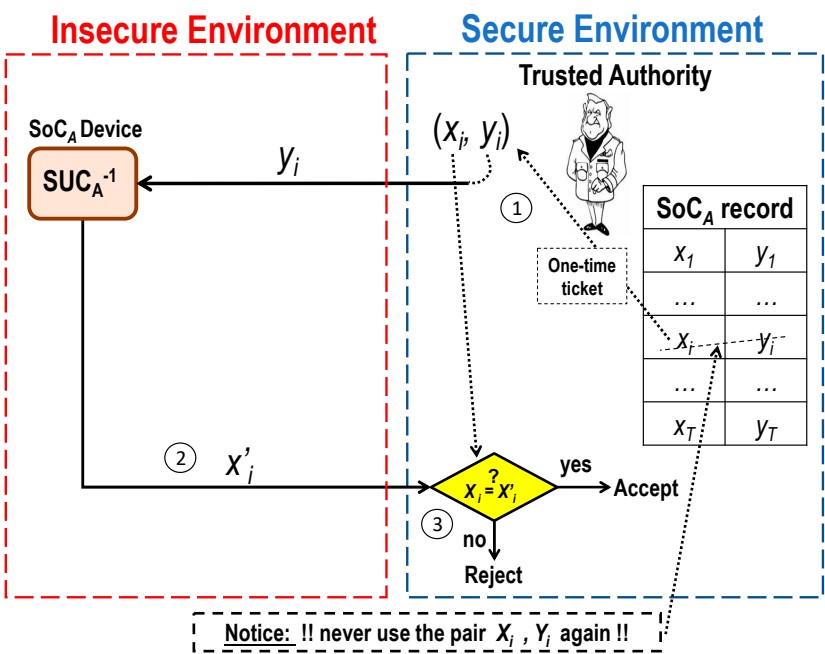

**Figure 5.** Two-way identification protocol over an insecure channel.

An SUC-based identification protocol may proceed as follows:

1.  A secret pair $(x_i, y_i)$ is randomly chosen from the TA's secret records of $SoC_A$. Then, the TA challenges the $SoC_A$ device by the cryptogram $y_i$ over an insecure channel.
2.  The $SoC_A$ device responds by sending the decrypted cleartext $x'_i$.
3.  If $x'_i = x_i$, then the $SoC_A$ device is deemed to be authentic, and the pair $(x_i, y_i)$ is then marked as a used pair and never used again avoiding replay attack for highest security.

*3.2. Modeling Attacks and Clone-Resistance Measures*

Machine learning (ML) can be deployed to create a predictive model of an unknown function, algorithm, and/or concept by analyzing some training data [24]. Such a learning approach can be used for cryptanalysis [25], especially for modeling attacks on PUFs [5]. In a special case, if a learner $L$ can predict the output of a PRF such as $f$ based on past training data such as $(x_1 f(x_1)), \dots, (x_q, f(x_q))$, then $L$ can be used to distinguish the output of this PRF $f$ [24], and $f$ is not a secure PRF.

A secure PRF concept postulates that the output of PRF is statistically independent of the training data and uncorrelated with any learner [24]. Therefore, if a designed SUC is a secure PRF, then there is no ML algorithm that can build a predictive model for such an SUC. In this case, the SUC is a modeling-resistant structure.

On the other hand, cloning an entity indicates the ability of reproducing the same entity. The unclonability of an SUC comes from the fact that nobody knows its structure. The important issue that the cipher designer faces is how to generate a cipher that the designer himself cannot predict.

The cloning-resistance entropy ($H_{CRE}$) for an SUC is proportional to the number $\sigma$ of all possible choices of a randomly selected SUC, so that $H_{CRE}$ is defined as:

$$H_{CRE}(SUC) = \log_2(\sigma). \tag{10}$$

If $H_{CRE}$ is a significant cryptographically large value, then the proposed SUC is claimed to be cloning resistance. The SUC design proposal is targeting $H_{CRE} > 500$ bits; that is, the cloning complexity is larger than $2^{500}$ cycles and/or memory bits.

In the following sections, a cipher creation strategy deploying modern VLSI devices as non-volatile FPGAs is presented. The key objective of this work is to use existing FPGA resources in an efficient way for creating very large classes of cipher structures and particularly by using the existing hard cores of arithmetic mathematical blocks called (Mathblocks). Such blocks are capable of multiplying and adding what would be the basic building blocks of the proposed SUC cipher structures to come up with low-cost realization possibilities by consuming available structures.

## 4. New SUC Implementation Strategy and Target FPGA Environment

The only non-volatile flash-based FPGA technology with programmable cells is available via Microsemi Smart-Fusion®2 devices. Some of the main features of the Smart-Fusion®2 FPGAs are flash-based fabric cells, a microcontroller subsystem based on an ARM Cortex-M3 processor, and high-speed hard cores of arithmetic Mathblocks called MACCs, including multipliers and adders [26].

The integrated MACCs are optimized to efficiently perform a DOT product mode as a $9 \times 9$ ($8 \times 8$ unsigned integers) multiplication and a normal mode as an $18 \times 18$ ($17 \times 17$ unsigned integers) multiplication, as shown in Figures 6 and 7, respectively.

## Block Diagram of the Math Block in DOT Product Mode

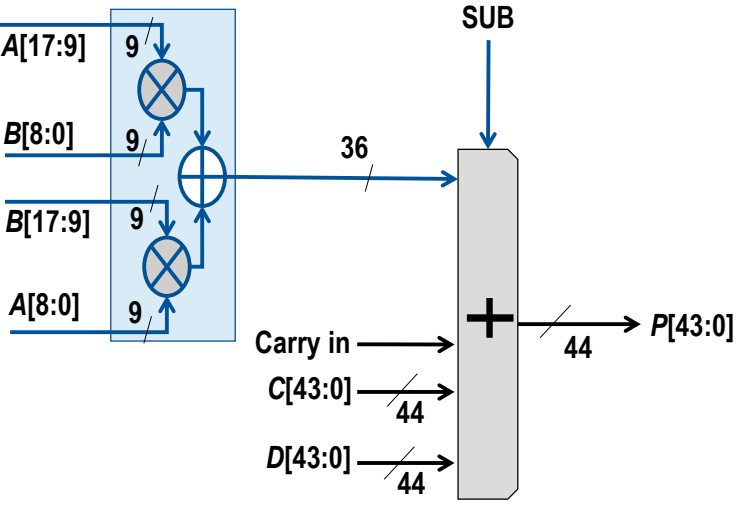

$$P[43:0] = (A[8:0] \times B[17:9] + A[17:9] \times B[8:0]) + C[43:0] + D[43:0] + Carry\ in$$

**Figure 6.** A DOT product mode of SmartFusion®2 FPGA using MACCs [26]. MACCs: high-speed hard cores of arithmetic Mathblocks [26].

## Functional Block Diagram of the Math Block in Normal Mode

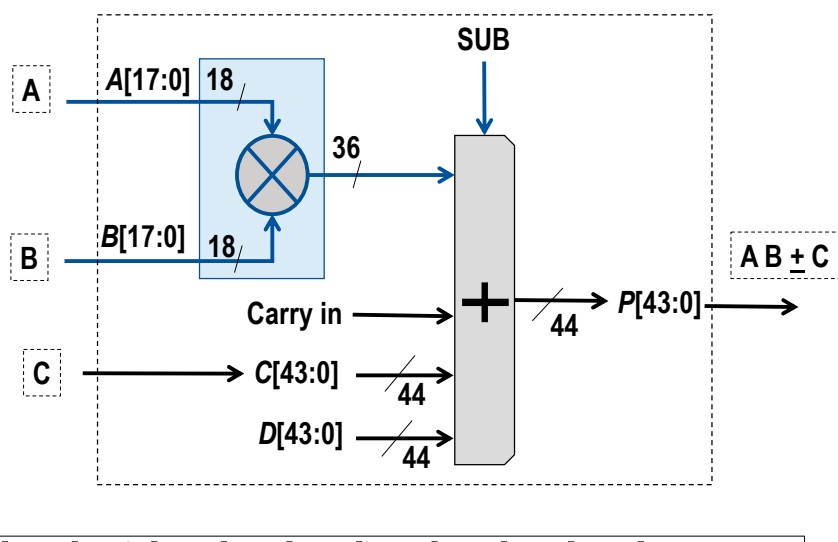

$$\boxed{P[43{:}0] = (A[17{:}0] \times B[17{:}0]) + C[43{:}0] + D[43{:}0] + \textit{Carry in}}$$

**Figure 7.** The constellation of multiply and add (AB+C) in Smart-Fusion®2 FPGA using MACC [26].

The new proposed implementation strategy is specially characterized for mainly using the following building blocks:

- The hard-core FPGA multiplier modules as shown in Figure 7 should be deployed as a backbone of the designed ciphers. The major novelty of the designed ciphers lies in using such hard-core multipliers (in normal mode) in the involution function, which includes both confusion and diffusion properties at the same time.
- The 4-bit look-up tables (LUT) cells should be used as small so-called Golden S-Boxes as lightweight nonlinear mappings having adequate security properties where each Golden S-Box requires just $4 \times 4$-input LUTs [26].

**The reason for this special building blocks selection:**

This implementation strategy is fully unusual in standard cipher designs as multipliers result with very high hardware complexity. However, in such FPGA technology, plenty of such multipliers may not be used in many applications and are left as dead entities. Deploying unused/dead modules for creating SUC structures is considered as a value creating a *"reanimation process"* of dead entity in this very special case.

The other DOT product mode of Figure 6 is also an objective of the author in ongoing research, which is outside of the scope of this paper. Figure 8 illustrates a possible functional layout after generating an SUC in a FPGA that uses MACCs interacting with some logical components LUTs consumed from the FPGA fabric and a minor software service program. This combination of MACCs and logic implemented components results in the desired target SUC of this proposal.

The ultimate security level of the SUC in FPGA technology is attained if the cipher locations in the layout are random and unknown. Note that the random and individual location of each SUC minimizes the risk of physical attacks considerably. This is even true when the adversary tries to obtain information by probing points inside the chip [27]. Random unknown allocating of the SUC structures physically in the FPGA layout is the subject of ongoing research and is outside the scope of this work.

**Sample SmartFusion2 SoC FPGA Structure Device**

**Figure 8.** A sample layout of an SoC unit after SUC creation [13].

## 5. A New Feistel-Like Cipher Class

A new design strategy for a Luby–Rackoff cipher is presented below by replacing the XOR operation with a new powerful self-inverse mapping (Latin Square). The proposed mappings design is based on deploying the MACCs in the ring of integers modulo $2^n$ in the SmartFusion®2 FPGA technology.

Several block ciphers were classified as a Feistel cipher [28] such as the data encryption standard (DES) [29], Camellia [30], LBlock [31], etc. In [32], Biham and Shamir replaced some of the XOR operations in DES by the addition of mod $2^n$. The resulting cipher becomes more resistant against differential cryptanalysis. A new construction of the Luby–Rackoff cipher as $\psi(h, f, f, h)$ was presented in [33], where $f$ is a PRF, and $h$ is a universal hash function. The resulting cipher structure $\psi(h, f, f, h)$ uses addition mod $2^n$ instead of XOR operation. This work is inspired by the following fact from [34]: "let $X = 2^n-1$, $Y = 1$ be the integer representation of a two *n*-bit block. In this case, the $X+Y$ mod $2^n$ is equal to zero, where all bits in $X$ are changed in the cryptogram, whereas $X \oplus Y$ is equal to $n-1$ ones and a zero in the last significant bit, which means, only the last significant bit is changed [34]". These results show that the same level of security between an XOR-based Luby–Rackoff cipher and the addition of a mod $2^n$ -based Luby–Rackoff [34] is attainable.

The first step toward replacing the XOR operation of DES by another operation was taken in [35]. The proposed cipher is constructed based on * operation, which is defined as a Latin square. Following this work, a new design for a Luby–Rackoff cipher is proposed by replacing the * (non-involutive) operation with a new MACC-based self-inverse (involutive) mapping. The resulting new cipher class is usable for self-created SUCs.

### 5.1. New Latin Square as Involution Mapping for SUC

In 2003, Klimov and Shamir [36] introduced a new class of low-complexity functions which are invertible and exhibit special properties. Such functions were called Triangular-Functions (T-function). A T-function is defined as follows:

**Definition 2.** *A function f(x) is called a T-function if the n-bit output of the function holds that the i-th bit of its output depends only on the first, the second, . . . , and the i-th bit of its inputs.*

Eight basic possible constructing operations of T-functions were introduced in [36] as:

- Negation (*-a*) mod $2^n$, Addition (*a+b*) mod $2^n$, Subtraction (*a-b*) mod $2^n$, and Multiplication (*a.b*) mod $2^n$.

- The Boolean functions; Complement $\bar{a}$, OR $(a \vee b)$, AND $(a \wedge b)$, and XOR $(a \oplus b)$.

where a and b are two n-bit words.

In [36], Klimov generalized Rivest's construction of permutation polynomials (PPs) [37] resulting with invertible mappings with T-functions properties as follows:

**Theorem 1 [36].** *Let* $P(x) = a_0 \overset{\pm}{\underset{\oplus}{}} a_1 x \overset{\pm}{\underset{\oplus}{}} \cdots \overset{\pm}{\underset{\oplus}{}} a_d x^d$ *be a generalized polynomial with integral coefficients. Then, $P(x)$ defines a permutation polynomial modulo $2^n$: $n > 2$, if and only if $a_1$ is odd, $(a_2 + a_4 + \cdots)$ is even, and $(a_3 + a_5 + \cdots)$ is even.*

Let $\Pi^2$ denote the set of all polynomials $P : \mathbb{Z}_{2^n} \times \mathbb{Z}_{2^n} \to \mathbb{Z}_{2^n}$ of two variables of degree 1 in the form:

$$P(L, R) = a.L \overset{\pm}{\underset{\oplus}{}} b.R \tag{11}$$

where $a, b, c \in \mathbb{Z}_{2^n}$. In this case, any polynomial $P$ from $\Pi^2$ is defined as a mapping having two inputs such as $(L, R)$ and one output $P(L, R)$ in $\mathbb{Z}_{2^n}$.

**Definition 3 (Latin Square) [37].** *The polynomial P(L,R) defined in Equation (11) over the ring $\mathbb{Z}_{2^n}$ is a Latin square if both functions $P(L, C)$ and $P(C, R)$ are permutations over $\mathbb{Z}_{2^n}$, for any $C \in \mathbb{Z}_{2^n}$.*

The following theorem determines the main requirements on $P(L, R) = aL + bR$ to become a Latin square over $\mathbb{Z}_{2^n}$.

**Theorem 2.** *Let $n > 1$ and $P(L, R) = a.L \overset{\pm}{\underset{\oplus}{}} b.R$ be a polynomial in two variables (L,R) over $\mathbb{Z}_{2^n}$. Then, P is a Latin square, if a and b are odd numbers.*

**Proof.** According to Theorem 1, $P(L, C) = aL \overset{\pm}{\underset{\oplus}{}} bC$ and $P(C, R) = aC \overset{\pm}{\underset{\oplus}{}} bR$ are permutations over $\mathbb{Z}_{2^n}$ if $a$ and $b$ are odd numbers. This implies that $P(L, R) = aL \overset{\pm}{\underset{\oplus}{}} bR$ is a Latin square based on Definition 3. $\square$

**Definition 4.** *The polynomial $P(L, R)$ on the ring $\mathbb{Z}_{2^n}$ is a self-inverse mapping with respect to L if:*

$$P(P(L, R), R) \bmod 2^n = L \tag{12}$$

*holds true for every L and R.*

The following theorem determines the main requirements on a Latin square $P(L, R) = aL \overset{\pm}{\underset{\oplus}{}} bR$ to reach such self-inverse mapping in two variables $(L, R)$ with respect to $L$ over $\mathbb{Z}_{2^n}$.

**Theorem 3.** *Let $n > 1$ and $P(L, R) = aL \overset{\pm}{\underset{\oplus}{}} bR$ be a Latin square with $a, b$ odd coefficients over $\mathbb{Z}_{2^n}$. P is a self-inverse mapping with respect to L if $a = \underbrace{1 \cdots 1}_{n} = 2^n - 1$.*

**Proof.** First, if $a = \underbrace{1 \cdots 1}_{n} = 2^n - 1$, then $a^2 = 1$ over $\mathbb{Z}_{2^n}$. In the following, it is proven that $(L, R) = aL + bR$ is a self-inverse Latin square with respect to $L$. The other cases of "$\oplus$" and "$-$" can be proven similarly.

Let,

$$P(P(L, R), R) = a.(a.L + b.R) + b.R.$$

And,

$$P(P(L, R), R) = a^2 L + b(a + 1)R.$$

Now,

$$P(P(L, R), R) = a^2 L + b(2^n - 1 + 1)R.$$

Yielding,

$$P(P(L, R), R) = a^2 L + 2^n bR.$$

Applying mod $2^n$ results with $P(P(L, R), R) \bmod 2^n = L$      □

Let $\Pi_i$ denote special classes of self-inverse mappings with respect to $L$ from $\Pi^2$; for $i = 1, 2, 3$, as follows:

$$\Pi_1 : P(L, R) = aL + bR, \; \Pi_2 : P(L, R) = aL - bR, \text{ and } \Pi_3 : P(L, R) = aL \oplus bR. \tag{13}$$

**Lemma 4.** *Let $n > 1$ and $P(L, R) = aL \overset{\pm}{\underset{\oplus}{}} bR$ be a Latin square with $a, b$ odd coefficients over $\mathbb{Z}_{2^n}$. $P$ is a self-inverse mapping with respect to $R$ if $b = \underbrace{1 \cdots 1}_{n}$.*

The resulting classes of self-inverse Latin squares (SILS) $\Pi_i : P(L, R)$ defined and so-called $\pi_i$ mappings are shown in Figure 9, where $\pi_i(L) = \left( aL \overset{\pm}{\underset{\oplus}{}} bR \right) \bmod 2^n$; $i = 1, 2, 3$. In the following, the number of all the distinct self-inverse polynomials with respect to $L$ is determined:

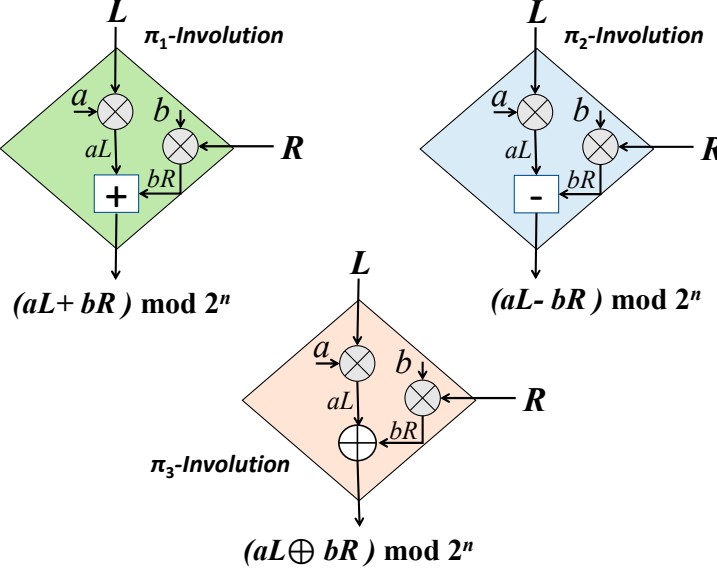

**Figure 9.** The new $\pi_i$ mappings used as involutions.

**Corollary 5.** *For $n > 3$, the cardinality of the class of all possible $\zeta$ over $\mathbb{Z}_{2^n}$ is $|\Pi_i| = 2^{n-1}$; for $i = 1, 2, 3$.*

**Proof.** From Theorem 3, the following is true:

$a = \underbrace{1 \cdots 1}_{n}$. That implies $|a| = 1$ and $b$ is odd, so $|b| = 2^{n-1}$.

Therefore, $|\Pi_i| = |a|.|b| = 1.2^{n-1} = 2^{n-1}$.　　　　□

It can also simply proven that the $\pi_i$ mappings are involutions for any $R$, as shown in Figure 10. Throughout the following sections, the focus lies on the class of mapping $\pi_1(L) = aL + bR$. The other cases of "$\oplus$" and "$-$" can be similarly investigated.

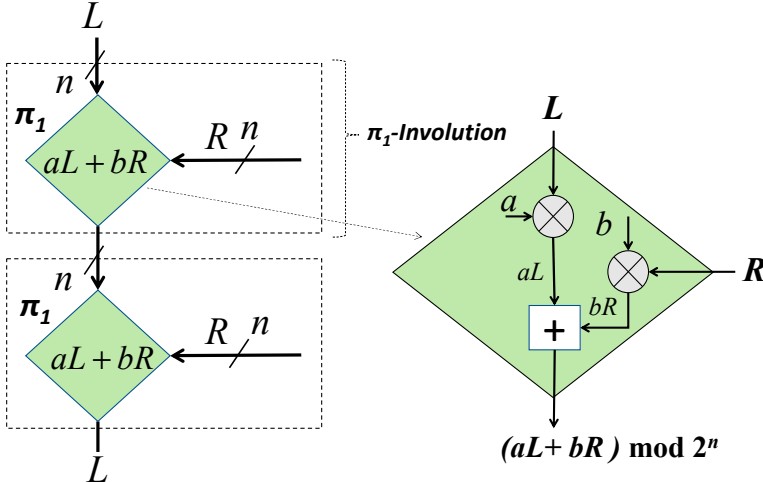

**Figure 10.** Proof that $\pi_1$ is an involution [13].

In a further round construction, when replacing the XOR operation in a Luby–Rackoff cipher with $P(L, R) = aL + bR$ results in a new mapping defined as follows:

$$\zeta(f)(L, R) = (aL + b_i f(R), R), \text{ for } i = 1, \cdots, 2^{n-1}. \tag{14}$$

It can be simply proven that $\zeta(f)(L, R)$ is also an involution for any $(L, R)$. Figure 11 shows the core mapping $\zeta$ of the new proposed Feistel-like cipher. The statistical properties of the multiplication and addition ensure that all the input bits will be affected (diffusion). Moreover, this construction is low-cost, since the SmartFusion®2 FPGA contains specific MACCs that are often readily available as unused components.

Figure 12 illustrates a proposed Feistel-like extended cipher round structure $\eta$. The round's input data is $2n$-bits, which splits into two branches of $n$-bits ($L$: left and $R$: right). Then, $\zeta$-involution is applied on both branches $(L, R)$, where the inner function $f$ is applied only on $R$.

Then, the round structure includes two mappings, namely: $\zeta$ involution followed by a swap involution mapping. The $t$-rounds of the proposed ciphers are using the same two involutions in each round with different $b$ parameters, which can be seen as round keys.

Note that the total number $\sigma$ of all possible constructible ciphers as an SUC class as Feistel-like ciphers $\eta$ having $t$ rounds depends generally on the total number $\mu$ of all possible inner functions $f$, where $\mu = 2^{n2^n}$.

$$\sigma = \max_{\mu}\{(2^{n-1})^t \mu\} = (2^{n-1})^t 2^{n2^n} = 2^{t(n-1)+n2^n} \tag{15}$$

The main advantage of the described "involutive" cipher structure is that the same function can be used for both encryption and decryption operations, differing only in using the keys in a reverse order.

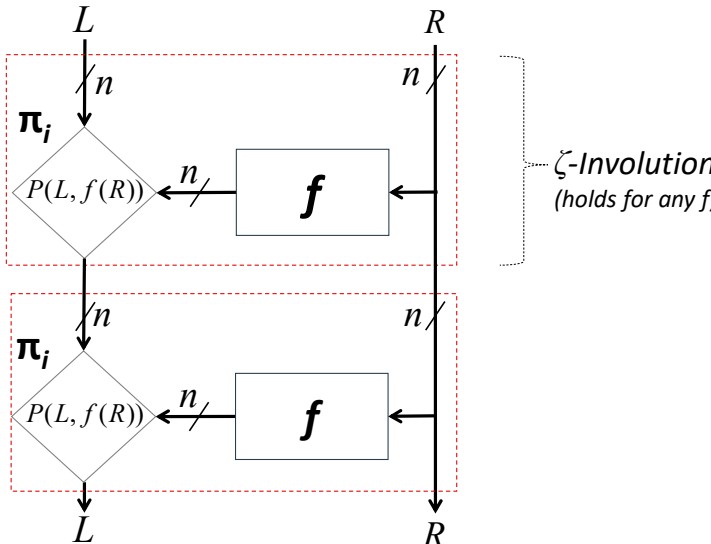

**Figure 11.** The new $\zeta$ involution for a cipher round structure. Adapted from [13].

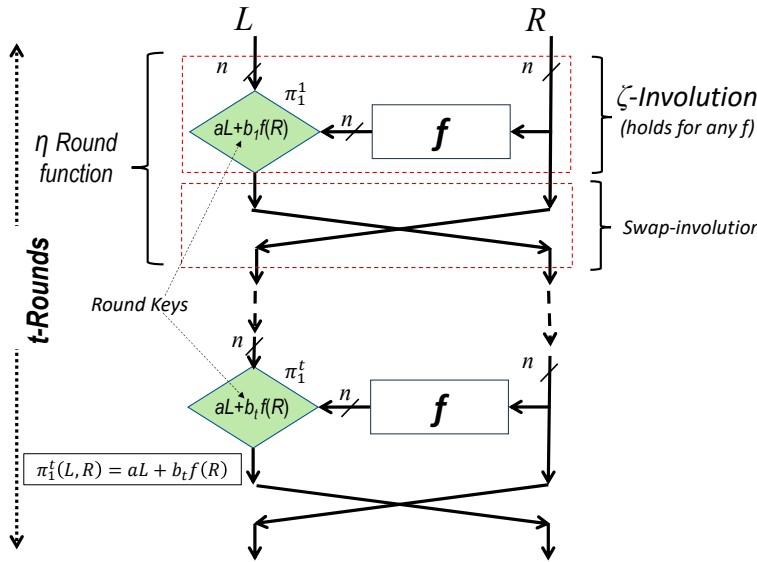

**Figure 12.** The proposed Feistel-like cipher and its round structure $\eta$.

### 5.2. Distinguishing Attack on the Proposed Feistel-Like Cipher

Let $B_{2n}$ ($\mathcal{F}_{2n}$) denote a set of all possible permutations (functions) from $\{0,1\}^{2n}$ to $\{0,1\}^{2n}$, where the cardinality of $B_{2n}$ ($\mathcal{F}_{2n}$) is $|B_{2n}| = 2^{2n}!$ ($|\mathcal{F}_{2n}| = 2^{n2^n}$), respectively. Furthermore, let $f \overset{U}{\leftarrow} \mathcal{F}$ randomly choose the function $f$ from $\mathcal{F}$ according to a probability uniform distribution $U$ over $\mathcal{F}$ where $\mathcal{F}$ is the set of all possible functions from $\{0,1\}^n$ to $\{0,1\}^n$.

The proposed Feistel-like cipher $\eta$ is defined as a permutation of $B_{2n}$ on a pair $(L_i, R_i)$ from $\{0,1\}^n \times \{0,1\}^n$ where $i = 1, 2, \ldots q$. The evaluation of the distinguishing attack on the proposed Feistel-like cipher is carried out by deploying the core mapping $\zeta$ as a mapping in different ciphering configurations. These structures can be developed based on distinguishing attack scenarios. The generic attacks [38] on one, two, and three identical rounds of the proposed Feistel-like cipher $\eta$ is explained in "Appendix A". The results show that none of the $\eta(f)$, $\eta(f,f)$, and $\eta(f,f,f)$ are PRPs (Pseudorandom Permutations).

To design a PRP cascade from a single PRF, it is required to have at least three different rounds of $\eta$ using a single PRF $f \overset{U}{\leftarrow} \mathcal{F}$ with different parameters in the $\pi$ mapping to attain a structure that is indistinguishable from a truly random permutation. The new structure of a Feistel-like cipher $\eta(f, f, f)$ should consequently include at least three subsequent different $\pi$ mappings as follows:

$$\left. \begin{array}{l} X_i = aL_i + b_j f(R_i) \\ S_i = aR_i + b_k f(X_i) \\ T_i = aX_i + b_l f(S_i) \end{array} \right\}, \tag{16}$$

where $f \overset{U}{\leftarrow} \mathcal{F}$, and $b_j \neq b_k \neq b_l$, which are acting as different round keys.

Now, let $\eta_t(f)$ denote $\eta(\underbrace{f, \cdots, f}_{t})$ with $t$ different odd values of $b_i$; $i = 1, \cdots, 2^{n-1}$. In this case, to prove that $\eta_3(f)$ is a PRP, a distinguishing experiment should be applied on $\eta_3(f)$. To attain that goal the following two Lemmas need to be valid for $\eta_3(f)$:

**Lemma 6.** *For every function* $G : \left(\{0,1\}^{2n}\right)^q \rightarrow \{0, 1\}$ *and for any q pairs* $(L_i, R_i) \in \{0,1\}^n \times \{0,1\}^n$, *where* $i = 1, \cdots, q$.

$$\left| \Pr\left[ G\big(\eta_3(f)(L_1, R_1), \cdots, \eta_3(f)\big(L_q, R_q\big)\big) = 1 : f \overset{U}{\leftarrow} \mathcal{F} \right] - P_G \right| \leq \frac{q^2}{2^n} \tag{17}$$

*where* $f \overset{U}{\leftarrow} \mathcal{F}$ *and* $P_G$ *defined as:*

$$P_G = \frac{\#\{(x_1, \cdots, x_q) \in (\{0,1\}^{2n})^q : G(x_1, \cdots, x_q) = 1\}}{2^{2nq}}. \tag{18}$$

**Proof.** (See "Appendix B"). $\square$

**Lemma 7.** (PRF Switching Lemma [39])**:** *For a distinguishing experiment, let E be a block cipher defined over* $(K, X)$, *where,* $|X| = 2^{2n}$. *Consider an adversary (distinguisher)* $\Psi$ *that makes at most q queries to its challenger. Then,*

$$\left| Adv_{PRP}^E(\Psi) - Adv_{PRF}^E(\Psi) \right| \leq \frac{q^2}{2^{n+1}}. \tag{19}$$

**Distinguishing Experiment $\eta_3(f)$:**

Step 1: For the proposed Feistel-like cipher $\eta_3(f)$ defined over $(K, X)$, where $|X| = 2^{2n}$. Consider an adversary (distinguisher) $\Psi$ that interacts with a challenger acting as follows:

- The challenger randomly chooses one bit $b \overset{U}{\leftarrow} \{0, 1\}$.
- The challenger returns $P \overset{U}{\leftarrow} B_{2n}$, if $b = 1$ to $\Psi$; otherwise, it returns $P \leftarrow \eta_3(f)$, where $f \overset{U}{\leftarrow} \mathcal{F}$ within time $t$.

Step 2: The adversary $\Psi$ submits to a challenger a polynomial number of queries $(q)$ such as $(L_i, R_i)$, where $i = 1, \cdots, q$.

Step 3: The adversary terminates the experiment by returning $b'$.

According to Lemma 6, the advantage of $\Psi$ to distinguish between $\eta_3(f)$ and a random function is:

$$Adv_{PRF}^{\eta_3(f)}(\Psi) \leq \frac{q^2}{2^n}. \tag{20}$$

Now, the PRF Switching Lemma [39] (Lemma 7) stated that,

$$Adv_{PRP}^{\eta_3(f)}\left(\Psi\right) \le Adv_{PRF}^{\eta_3(f)}\left(\Psi\right) + \frac{q^2}{2^n}. \tag{21}$$

So that,

$$Adv_{PRP}^{\eta_3(f)}\left(\Psi\right) \le \frac{3q^2}{2^{n+1}}. \tag{22}$$

The last result in Equation (22) concerning the proposed Feistel-Like ciphers shows that it attains the same security bound as that of the Luby–Rackoff cipher.

## 6. New $\pi_i$-Mappings Hardware Structure and Its Complexity

In this section, the $\pi_i$ mappings are modeled and implemented in Microsemi Smart-Fusion®2 FPGA technology.

The hardware complexity of each implemented $\pi_i$ mapping was evaluated based on the number of consumed MACCs, LUTs (Look Up Tables), and DFFs (Delay Flip Flop), where the hardware realization of these mappings is fundamentally implemented based on using a wide multiplier of size larger than $18 \times 18$. Here, a wide multiplier is efficiently implemented by using a cascade of many MACCs [26]. The chosen FPGA from the Microsemi FPGAs family is a Smart-Fusion®2 M2S025, which contains 27,696 LUTs, 27,696 DFFs, and 34 MACCs. Figure 13 illustrates the resource utilization for $\pi_1$ mapping with an input size of $n = 17$ and 18 bits. The consumed resources of $\pi_1$ mapping with $n = 17$ bits are two MACCs and 17 LUTs.

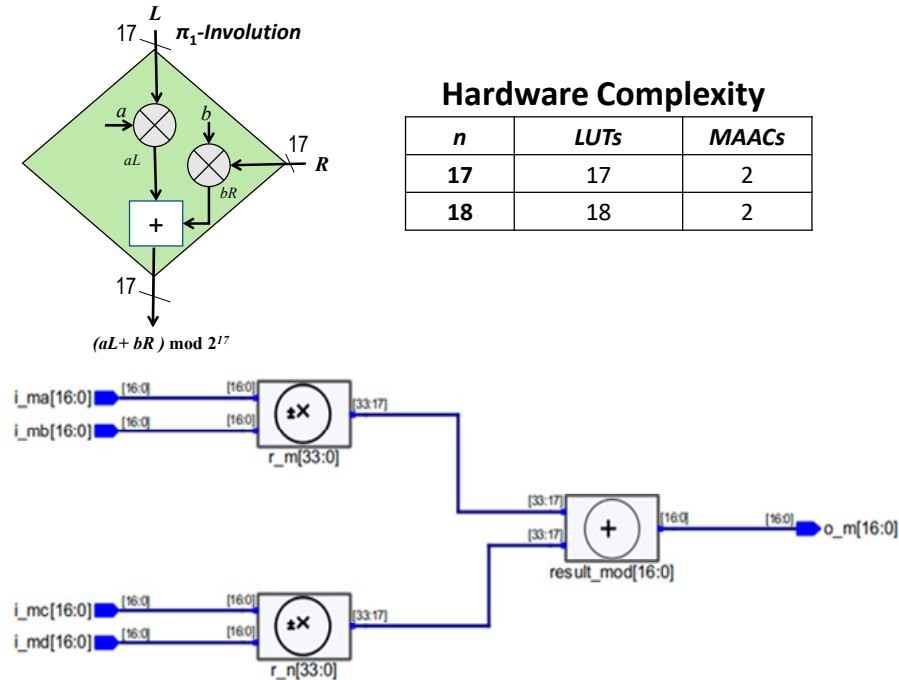

**Figure 13.** FPGA implementation of $\pi_1$ mapping for 17- and 18-bits as input size.

For $n = 32$ and 34 bits as input size, two wide multipliers were deployed. In this case, each wide multiplier is realized as a cascade of four MACCs. Figure 14 shows the required number of MACCs to build two wide multipliers consuming 32 LUTs for $n = 32$, and 34 LUTs for $n = 34$.

Figure 15 shows the required number of MACCs to implement $\pi_2$ and $\pi_3$ mappings with the input sizes of $n = 17$, 18, 32, and 34.

Note that two MACCs are required when $n = 17$ or 18, and two wide multipliers are implemented as a cascade of MACCs for $n = 32$ or 34.

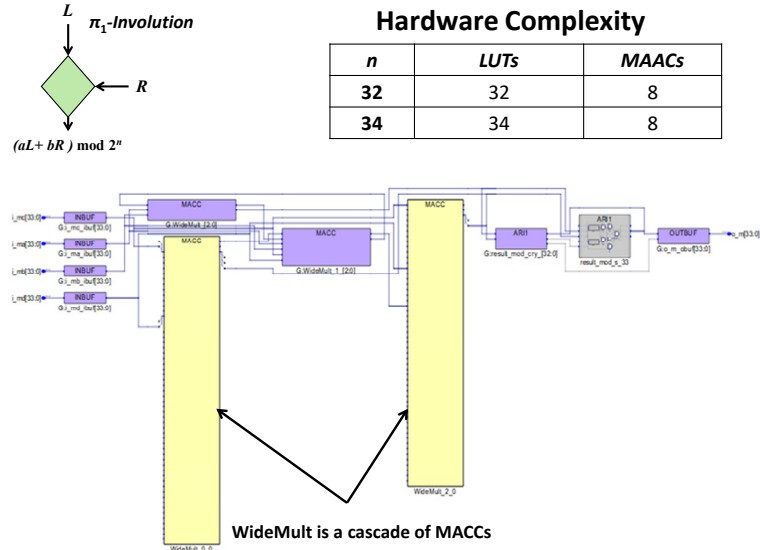

**Figure 14.** FPGA implementation $\pi_1$ mapping using two wide multipliers with input sizes of 32 bits and 34 bits.

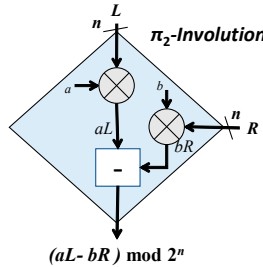

| n | LUTs | MAACs |
|---|------|-------|
| 17 | 18 | 2 |
| 18 | 19 | 2 |
| 32 | 33 | 8 |
| 34 | 35 | 8 |

$\pi_2$- *Mapping* **Hardware Complexity**

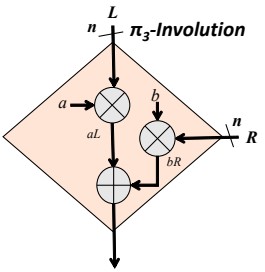

$\pi_3$- *Mapping* **Hardware Complexity**

| n | LUTs | MAACs |
|---|------|-------|
| 17 | 17 | 2 |
| 18 | 18 | 2 |
| 32 | 33 | 8 |
| 34 | 34 | 8 |

**Figure 15.** FPGA implementation of $\pi_2$ and $\pi_3$ mappings for input sizes $n = 17, 18, 32,$ and 34 bits.

## 7. Possible Feistel-Like Inner Function Design

The need for a simple low-cost implementation of the inner functions $f$ of the proposed cipher $\eta_t(f)$ led to deploy a huge class of cryptographically significant $f$ mappings. In this section, the necessary design strategy for the inner function $f$ with good cryptographic properties is presented.

### 7.1. Golden 4-Bit S-Boxes as Basic Building Elements for the Mapping of f

In [40], Saarinen showed that only four classes of S-Boxes (4-bit to 4-bit mapping) can affinely transform the resistance properties against linear cryptanalysis (LC) and differential cryptanalysis (DC) to all S-Box classes. These optimal 4-bit S-Boxes are called golden S-Boxes (GS). Moreover, a new equivalence relation is defined based on two bit permutation matrices $P_i, P_j$, two values $a, b \in \mathbb{F}_2^4$, and two XOR operations as follows,

$$[S(x)]_{1\times4} = GS_k(([(x)]_{1\times4} \oplus [a]_{1\times4}) \cdot [P_i]_{4\times4}) \cdot [P_j]_{4\times4} \oplus [b]_{1\times4} \tag{23}$$

where $GS_j$ is a GS for $k = 0, 1, 2, 3$ (see Table 1) and $x \in \mathbb{F}_2^4$. Table 1 shows the four GS seeds that satisfy the ideal properties for all class members [40].

**Table 1.** Four golden S-Boxes (GS) seeds for S-Box generators [40]. DC: differential cryptanalysis, LC: linear cryptanalysis.

| GS Seed-Classes | 4-Bit Input Combinations 0123456789ABCDEF | DC $p$ | LC $\varepsilon$ |
|---|---|---|---|
| $GS_0$: 4-bit outputs | 035869C7DAE41FB2 | $\frac{1}{4}$ | $\frac{1}{4}$ |
| $GS_1$: 4-bit outputs | 03586CB79EADF214 | $\frac{1}{4}$ | $\frac{1}{4}$ |
| $GS_2$: 4-bit outputs | 03586AF4ED9217CB | $\frac{1}{4}$ | $\frac{1}{4}$ |
| $GS_3$: 4-bit outputs | 03586CB7A49EF12D | $\frac{1}{4}$ | $\frac{1}{4}$ |

$\varepsilon$: linear probability bias, $p$: differential characteristics probability.

The cardinality of the class of all possible such GSs is then,

$$4.\left(2^4\right)^2 . (4!)^2 = 2^{19.1} \text{ Different GSs} \tag{24}$$

where 4 is the number of GS seeds, $GS_j$, $(2^4)^2$ is the number of all possible a, b parameter choices $a, b \in \mathbb{F}_2^4$, and $(4!)^2$ is the number of all possible bit permutation matrices $P_i$, $P_j$.

Figure 16 shows possible hardware mapping blocks for the GS generator according to Equation (23). The resulting generated S-Boxes exhibit equivalent cryptographic security performance.

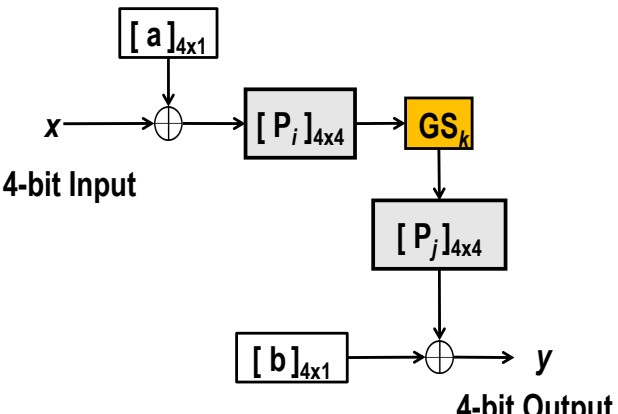

Implementation according to (23)

**Figure 16.** Hardware structure for the golden S-Box generator.

*7.2. Bricklayer Function as a Possible Inner Function f*

One of the simplest architectures of the inner function of the proposed Feistel-like cipher can be considered as a bricklayer function [41]. Here, the proposed bricklayer function can be seen as a Boolean function that is composed of parallel components or GSs of smaller inputs [41]. As the currently known GSs have an input/output size of $4 \times 4$ bits, only 32 bits are used for the inner function, since the maximum size of 34 is not divisible by 4. To make use of the full 34 bits, a further design adaptation is required. This is the objective of future research. In this case, the proposed bricklayer function is simply constructed as shown in Figure 17 and mathematically defined as:

$$f(x) = (GS_1(x_1), \cdots, GS_8(x_8)) \tag{25}$$

where $x = (x_1, \cdots, x_8)$ and $x_i \in \{0, 1\}^4$: for every $i > 0$.

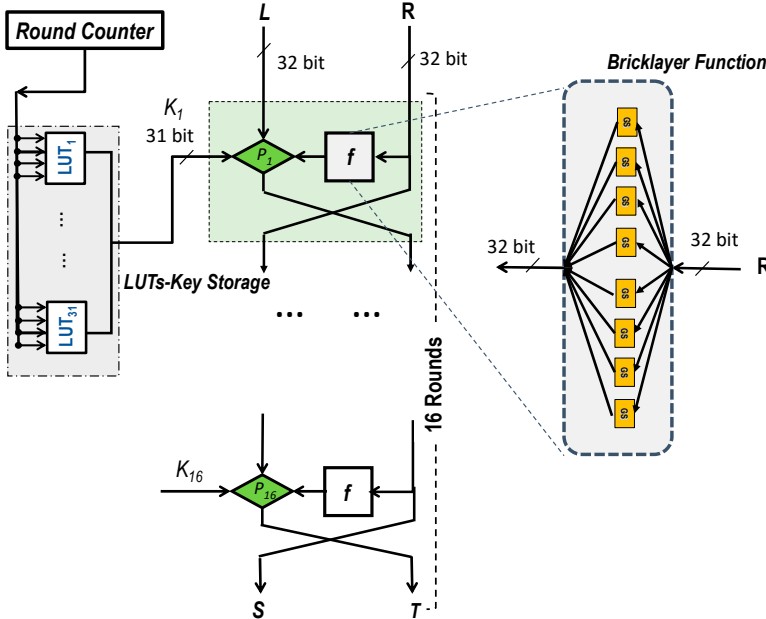

**Figure 17.** Possible design of the proposed cipher based on the Bricklayer function using GSs.

### 7.3. How Can the SUC-Creating GENIE Work?

The SUC creation process is performed by a GENIE program that will run in an enrollment process for each device. Assume that the GENIE would realize a Feistel-like cipher with a bricklayer function as an inner function delineated in Figure 17, where the input data size is $2n = 64$.

The GENIE may randomly create the proposed cipher as follows:

- The GS generator according to Equation (23) requires 128 storage bits for each GS seed and 16 storage bits for each possible bit permutation matrix. Therefore, the GS generator requires a total of $4 \times 128 + 24 \times 16 = 896$ storage bits for the four GS seeds and 24 possible bit permutation matrices.
- The GENIE generates randomly eight GSs for $f$ by randomly selecting all the parameters of Equation (23) through the TRNG output bit stream. Note that according to Equation (23), the GENIE consumes $20 \times 8$ GSs = 160 TRNG bits to create all eight GSs where each generated GS requires 20 bits, namely: 2 bits for selecting one GS seed out of four GS seeds, $2 \times 4 = 8$ bits for selecting the parameters $a, b \in \mathbb{F}_2^4$, and $2 \times 5 = 10$ bits for selecting the two permutation matrices $P_i$, $P_j$ out of all 24 permutation matrices.
- The GENIE consumes additionally $31 \times 16$ rounds = 496 TRNG bits for all 16 round keys to be stored in 31 LUTs. A round key is the 31-bits $b_i$ parameter in the mapping $aL + b_iR$; in each round $i$ for $i = 1 \dots ,16$.
- When the GENIE completes the cipher creation, the GENIE deletes itself fully and irreversibly.

**Overall GENIE complexity:** A total of $496 + 160 = 656$ TRNG bits and 896 memory storage bits in addition to about 18 instruction cycles are needed to create a single cipher choice.

Notice that the total number $\sigma$ of all possible SUCs of the proposed Feistel-like ciphers $\eta$ with *16*-rounds is

$$\sigma = 2^{31 \times 16} \times 2^{8 \times 19.1} \approx 2^{649}. \tag{26}$$

### 7.4. A Possible Prototype Hardware Implementation

To implement one of the possible compact versions of the proposed cipher with an input size of $2n = 64$ bits, the architecture of Figure 18 is proposed as a recursive round-based implementation [42].

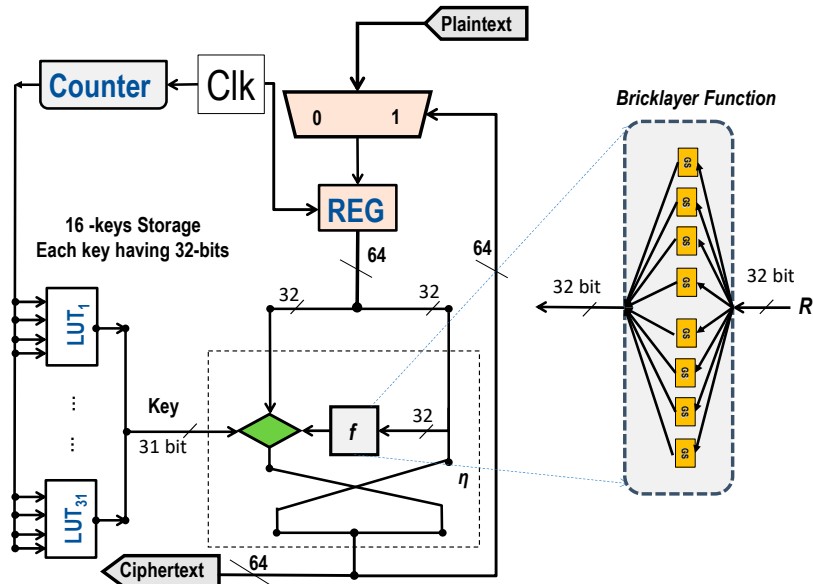

**Figure 18.** Proposed hardware architecture of the proposed SUC.

The aim of the designed structure is to iterate one cipher round $\eta(f)$, where a state machine is deployed to run the 16 cipher rounds using a state register of 64 bits and a 64-bit multiplexer, in which each cipher round is executed in one clock cycle.

Furthermore, a new technique of key scheduling was presented in [13] storing the 16 round keys in 31 LUTs, as shown in Figure 18. The keys are arbitrarily and randomly chosen by the GENIE.

Table 2 shows the resulting hardware complexity of the sample proposed SUC implementation in a SmartFusion®2 SoC FPGA. Further, more optimized implementations are under investigation.

**Table 2.** Hardware complexity using SmartFusion®2 M2S025T FPGA.

| Hardware Resources | LUTs | | DFFs | | MACCs | |
|---|---|---|---|---|---|---|
| | #LUTs | % | #DFFs | % | #MACCs | #All available MACCs |
| SUC using $\pi_1$ | 174 | 0.62 | 70 | 0.25 | 8 | 34 |
| SUC using $\pi_2$ | 175 | 0.63 | 70 | 0.25 | 8 | 34 |
| SUC using $\pi_3$ | 175 | 0.63 | 70 | 0.25 | 8 | 34 |

The implementation is aiming to evaluate the hardware complexity. A real implementation procedure is currently not possible, as Microsemi does not allow self-reconfiguration in its current devices. This is expected in future device generations.

## 8. Security Analysis and Evaluation

In this section, modeling attacks on the proposed SUC are discussed and the security level of the proposed SUC is evaluated by using the cryptanalysis of a cipher with secret components. Then, a quantum exhaustive search for SUC-Model is presented.

### 8.1. Modeling Attacks on SUC

In modeling attacks, the adversary tries to construct an ML algorithm that behaves indistinguishably from the original function (such as PUF) on almost all CRPs [5]. According to Section 5.2, the proposed SUC is a secure PRF. This implies that the output of the SUC is statistically independent of $(x_1, SUC(x_1)), \ldots, (x_q, SUC(x_q))$ and uncorrelated with any learner. Therefore, there is no ML algorithm that can build a predictive model for such SUCs.

*8.2. Cryptanalysis of a Cipher with Secret Components*

For more practical analysis, we identify an SUC from $\sigma = 2^{649}$ different SUCs in Equation (26), where the cipher input size is $2n = 64$ bits. Thus, the successful prediction of the adversary is possible with a probability,

$$\frac{1}{(2^{19.1})^8 (2^{31})^{16}} \approx \frac{1}{2^{649}}. \tag{27}$$

In [43], an attack on a block cipher with secret components analyzes only the known plaintext–ciphertext pairs attack to recover the secret cipher components one by one. According to this attack scenario and assuming that the adversary tries to attack one SUC as a Feistel-like cipher, the worst-case scenario is when only GSs are unknown components ignoring the round keys, as they may be reachable. It is also assumed that the adversary knows the parameters of the SUC without being able to access the round's inputs and outputs.

The adversary starts by gathering selected $T$ pairs of plaintexts of the form [43],

$$P_{L,r} = \left\{ (L_i, x \| r_j); x \in \mathbb{F}_2^4 \right\} \tag{28}$$

where $L_i \in \mathbb{F}_2^{32}$, and $r_j \in \mathbb{F}_2^{28}$; for $0 \le i, j \le T$. After that, the adversary finds all pairs $\{x, y\}$ from $P_{L,r}$ such that:

$$(L_i, x \| r_j) \oplus (L_i, y \| r_i) = (0^{32}, x \oplus y \| 0^{28}) \text{ for } 0 \le j \le T \tag{29}$$

where $0^k$ denotes the bit block of $k$ zeros. Then, the adversary determines the counter set $C(\{x, y\})$ based on the corresponding ciphertext differences of all $L_i, r_j$ as follows,

$$C(\{x, y\}) = \left\{ L_i, r_j \middle| \exists k;\ \eta_{16}(f)(L_i, x \| r_i) \oplus \eta_{16}(f)(L_i, y \| r_i) = \left( 0^{4k} \| e \| 0^{60-4k} \right) \right\} \tag{30}$$

where $e \in \mathbb{F}_2^4$. In order to recover only one GS, the adversary uses $C(\{x, y\})$ to count how often only one active GS is involved in the ciphertext difference, if the following is met,

$$e = GS(x) \oplus GS(y). \tag{31}$$

Let $D_e$ be the set of all $\{x, y\}$ pairs that hold Equation (31). According to [43], if the hamming weight $hw(e) = 1$, then finding four sets of form $D_e$ is enough to determine uniquely the targeted GS. Finding three sets of form $D_e$ determines eight possible S-Boxes as candidates, etc.

To evaluate this attack, the minimum number of active GSs in any differential trials through all 16 rounds is required to be found, where a differential trail is a sequence of the input and output differences in each round. This leads to following definition:

**Definition 5.** *In DC, an S-box is active in a differential trail if and only if its input difference is non-zero [44].*

An exhaustive search was performed for a sample of 20,000 different ciphers using the following properties:

- The right left subblock is $R = 0$ and the left subblock is $L = (L_1, L_2, L_3, L_4, L_5, L_6, L_7, L_8)$, where $L, R \in \mathbb{Z}_{2^{32}}$ and $L_j \in \mathbb{Z}_{2^4}$ for $j = 1, \ldots, 8$.
- Let $\Delta L_1 = \Delta x$ denote all possible differences, for all $x \in \mathbb{Z}_{2^4}$. Then, the input difference of a generated $f$ is $\Delta L = (\Delta x \| 0^{28})$.
- If the input difference of a $GS_i$ is non-zero, then the output difference will be non-zero.
- Applying $\zeta$ mapping on any zero-differential values will produce a zero-differential value.
- Applying $\zeta$-mapping on any non-zero-differential values will produce either a non-zero-differential value or a zero-differential value if it is a multiple of $2^4$.

Figure 19 illustrates an experimental security analysis on 20,000 randomly selected different ciphers from the proposed class to figure out the minimum number of differentially active GSs. It was found that after four rounds and for all 20,000 ciphers, at least 12 to 18 GSs (out of 32) were differentially active (shown as colored circles). After increasing the number of rounds, the active GSs increased proportionally. The ciphers having only 12 active GSs after four rounds mostly stayed in the bottom in their number of active boxes (marked as bold blue circles) but never diverted far away from the remaining sample ciphers. After 10 rounds, at least 48 GSs (out of 80) were active. The security analysis in made based on the worst case active GSs.

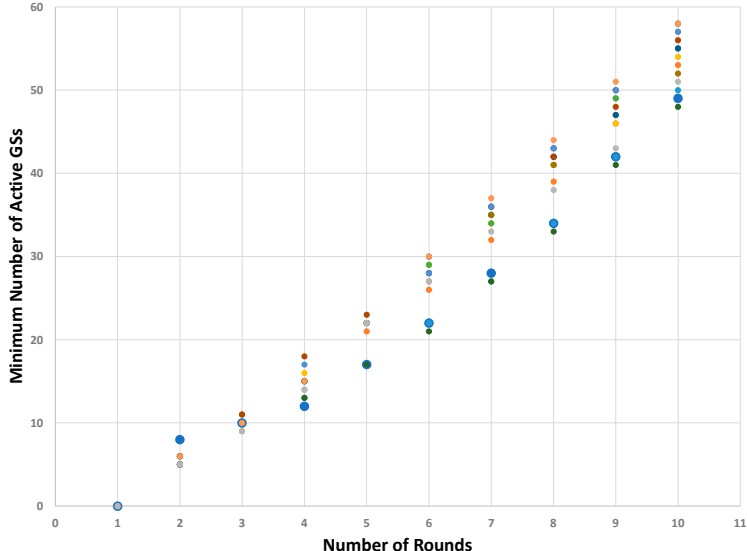

**Figure 19.** Minimum number of active GSs in a differential trial for 10 rounds of the ciphers sample.

Therefore, the probability $\Pr_i[\{x, y\}]$ that the ciphertext difference has $i$-th active GSs on the plaintext from $P_{L,r}$ in the worst case is,

$$\Pr_1[\{x, y\}] \leq \Pr_{12}[\{x, y\}] < \left(\left(\frac{1}{4}\right)^{12}\right)^4 = \frac{1}{2^{96}}. \tag{32}$$

Therefore, it can be concluded that the proposed SUC is sufficiently secure against any adversary who just analyzes the plaintext–ciphertext pairs.

### 8.3. Post-Quantum Exhaustive Search for SUC Model

In cryptography, Grover's algorithm is considered to be a special case of a more general search algorithm for quantum exhaustive searches [39]. For instance, Grover's algorithm finds $k_0$ from $K$ in $\sqrt{|K|}$ steps by querying a given function $\phi : K \rightarrow \{0, 1\}$ defined as:

$$\phi(k) = \begin{cases} 1; k = k_0 \\ 0; k \neq k_0 \end{cases}. \tag{33}$$

To identify an SUC from all generated Feistel-like ciphers by using Grover's algorithm, $\sqrt{\sigma}$ steps are required, which results practically with,

$$\sqrt{\sigma} \approx 2^{\frac{649}{2}} \approx 2^{325} \text{ steps.} \tag{34}$$

As a consequence, the proposed Feistel-like ciphers are even secure for post-quantum attacks.

### 9. Discussion and Conclusions

Designing a Secret Unknown Cipher Generator together with large adequate cipher classes is a very challenge task. The resulting SUC hardware complexity should be as low as possible to allow flexible and fast production. On the other hand, very large cipher classes are required with acceptable security quality, which may increase the complexity of the created cipher structure and its creating GENIE. The paper proposes a trade-off in an FPGA environment by "reanimating" certain unused arithmetic units to come up with acceptable practical hardware complexity. The new designed cipher class is restricted in its building block resources as it is using only certain hard-core arithmetic units.

The resulting complexities of the proposed designs are quite promising, consuming less than 1% of the device resources in one of the smallest SoC FPGAs. The proposed created SUCs are supposed to serve as a digital PUF alternative to the analog traditional PUFs. Our proposal exhibits attractive properties and is efficiently usable in emerging future IoT applications.

Furthermore, the security levels of the resulting SUC class are scalable. The proposed SUC as a Feistel-like cipher has a high level of security proportional to the hardware complexity. Moreover, the cipher design is a modified version of a well-investigated Luby–Rackoff cipher structures, which have been exposed to intensive review in the public literature. We expect to attain the same security bounds of the Luby–Rackoff cipher classes. As the cipher design is equivalent to a PRF in its design, there is no ML algorithm that can attack such SUCs as in the traditional PUFs. Finally, the resulting SUC's security level can cope easily with post-quantum security requirements as well by minor scaling on their complexity.

**In summary**, a new hardware-oriented cipher design for SUCs optimized for practical real-world environment is introduced.

Most FPGA applications do not consume all of the FPGA resources in particular powerful and complex multiplier cores. The ultimate goal of the SUC design is to embed the SUC in the FPGA without cutting resources from the functional FPGA duties. The reason is that embedding SUCs and personalization is processed at the very late stage by the end manufacturer before releasing the products to the market. This allows the end manufacturer to attain the highest security, as all the subcontractors would have no influence on the security management. In other words, the end manufacturer can easily produce his different components outside his factory without having any fear that the subcontractor would be able to clone his products, as subcontractors are fully out of the security process. Any produced component cannot be used without the SUC approval of the end manufacturer. Then, cloning by pirate companies or subcontractors is prohibited and the original product's royalties are fully protected.

**Zero-Cost Aspects**: The cipher design is deploying mainly hard-core (complex) multipliers as major building blocks, which may be available unused in modern system-on-chip (SoC) FPGA devices. The ultimate target of the cipher design is to allow "reanimating" spare unused multiplier cores to convert devices into clone-resistant units at possibly zero cost. Zero cost is assumed to be attained when embedding such an SUC module in a device does not consume any area cut from the usual application resources.

An ongoing research to devise many other new alternative implementations in future reconfigurable VLSI devices is in progress.

**Author Contributions:** Conceptualization, S.M. and W.A.; Methodology, S.M.; Software, S.M.; Supervision, W.A.; Validation, S.M.; Formal analysis, S.M.; Visualization, W.A.; Writing—original draft preparation, S.M.; Writing—review and editing, S.M. and W.A.; Project administration, W.A.

**Funding:** This work was supported by the DAAD Research Grants-Doctoral Programmes in Germany Nr. (57214224) and the German Federal Foreign Office scholarship funding (STIBET) program as well as Microsemi, a Microchip Company, San Jose, CA, USA and Volkswagen AG-Germany.

**Conflicts of Interest:** The authors declare no conflict of interest.

## Appendix A

The generic attacks on one, two, and three rounds of the proposed Feistel-like cipher follows the work of Patarin [38].

The one round Feistel-like cipher is described as:

$$(S_i, \ T_i) = \eta_1 \ (f) \ (L_i, \ R_i) = (P(L_i, \ f \ (R_i)), \ R_i) = (aL_i + bf(R_i), \ R_i). \tag{A1}$$

For one round:

$$\left. \begin{array}{l} S_i = R_i \\ T_i = aL_i + bf(R_i) \end{array} \right\}. \tag{A2}$$

The adversary can just test if $S_i = R_i$ for every $i$. This will happen with 100% probability after one query. Therefore, one round of the proposed Feistel-like cipher is not a PRP.

For two rounds:

The proposed Feistel-like cipher $\eta_2(f)(L_i, \ R_i)$ can be described as:

$$\left. \begin{array}{l} S_i = aL_i + b_1 f(R_i) \\ T_i = aR_i + b_2 f(S_i) \end{array} \right\}. \tag{A3}$$

If $b_1 = b_2$, the adversary chooses two pairs $(L_1, \ R_1)$ and $(L_2, \ R_2)$, where $R_1 = R_2$ and $L_1 \neq L_2$. Then, the adversary can just test if $S_1 - S_2 = a(L_1 - L_2)$. This will happen with 100% probability after four queries. Therefore, the proposed Feistel-like cipher with two rounds is not a PRP.

For three rounds:

The proposed Feistel-like cipher $\eta_3(f) \ (L_i, \ R_i)$ can be described as:

$$\left. \begin{array}{l} X_i = aL_i + bf(R_i) \\ S_i = aR_i + bf(S_i) \\ T_i = aX_i + bf(S_i) \end{array} \right\}. \tag{A4}$$

If $b_1 = b_2 = b_3$, the adversary will perform the following steps:

- Choose $(L_1, \ R_1) = (0, 0)$ as a query for $\eta_3(f)$ resulting with $(S_1, \ T_1)$.
- Choose $(L_2, \ R_2) = (0, S_1)$ as a query for $\eta_3(f)$ resulting with $(S_2, \ T_2)$.
- Choose $(L_3, \ R_3) = ( \ T_1 - aT_2, \ S_2)$ as a query for $\eta_3(f)$ resulting with $(S_3, \ T_3)$.

Then, the adversary can just test if $S_3 = aS_2 + S_1$ for $a = \underbrace{1 \cdots 1}_{n}$. This will happen with 100%

probability after at most $O(2^{n+1}) = O(2^{n+1} + 2^n)$ queries. Therefore, the proposed Feistel-like cipher with three identical rounds is not a PRP.

## Appendix B

The proof of Theorem 6 follows the framework of Maurer [45].

**Proof of Theoem 6.** Assume without loss of generality that the $q$ pairs $(L_i, \ R_i)$ are distinct. According to Equation (18), the outputs of the first, second, and third round are $(R_i, \ X_i)$, $(X_i, \ S_i)$, and $(S_i, \ T_i)$, respectively. Let $A_X$ be the event where $\{X_i\}_{i=1}^{q}$ is distinct, and let $A_S$ be the event where $\{S_i\}_{i=1}^{q}$ is distinct. Then, $A_X \cap A_S$ is the event where $\{X_i\}_{i=1}^{q}$ and $\{S_i\}_{i=1}^{q}$ are distinct.

Now, if the event $A_X$ occurs, then the values $S_i = aR_i + b_k f(X_i)$ are random for $i = 1, \ldots q$, where $b_k f(X_i)$ is a multiplication of two random values. On the other hand, $f \xleftarrow{U} \mathcal{F}$ and $b_l \xleftarrow{U} \{0, 1\}^{n-1}$; therefore, if the event $A_S$ occurs, then the values $T_i = aX_i + b_l f(S_i)$ are random for $i = 1, \ldots q$. In this case, $\eta_3(f)$ behaves precisely similar to a randomly chosen function from $\mathcal{F}_{2n}$, and the probability of distinguishing between $\eta_3(f)$ and a random function from $\mathcal{F}_{2n}$ is:

$$\left| \Pr\Big[ G\big(\eta_3(f)(L_1,R_1),\ \cdots,\eta_3(f)\big(L_q,R_q\big)\big) = 1 : f \overset{U}{\leftarrow} \mathcal{F} \Big] - P_G \right| \leq 1 - \Pr[A_X \cap A_S] \tag{A5}$$

and

$$1 - \Pr[A_X \cap A_S] = \overline{\Pr[A_X \cap A_S]} = \Pr\big[\overline{A_X} \cup \overline{A_S}\big] \leq \Pr\big[\overline{A_X}\big] + \Pr\big[\overline{A_S}\big] \tag{A6}$$

where $\overline{A_X}$ $(\overline{A_S})$ in the complementary event of $A_X$ $(A_S)$ occurring when $\{X_i\}_{i=1}^{q}$ $(\{S_i\}_{i=1}^{q})$ are not distinct, respectively.

For $i \neq j$, and according to the main assumption that the $q$ pairs $(L_i, R_i)$ are distinct,

$$\Pr\big[\overline{A_X}\big] = \binom{q}{2} \sum_{1 \leq i < jq} \Pr\big[X_i = X_j\big], \text{ and } \Pr\big[\overline{A_S}\big] = \binom{q}{2} \sum_{1 \leq i < jq} \Pr\big[S_i = S_j\big] \tag{A7}$$

where $\binom{q}{2}$ is the number of choosing two equal values $\big[X_i = X_j\big]$ $\big(\big[S_i = S_j\big]\big)$ out of $q$ from $\overline{A_X}$ $(\overline{A_S})$, respectively. On the other hand, the $q$ pairs $(L_i,\ R_i)$ are distinct by the assumption, and $\Pr\big[X_i = X_j\big]$ and $\Pr\big[S_i = S_j\big]$ are computed as,

$$\Pr\big[X_i = X_j\big] = \begin{cases} 2^{-n}; R_i \neq R_j \\ 0;\ R_i = R_j \end{cases}, \text{ and } \Pr\big[S_i = S_j\big] = \begin{cases} 2^{-n}; X_i \neq X_j \\ 0;\ X_i = X_j \end{cases}. \tag{A8}$$

From Equations (A7) and (A8),

$$\Pr\big[\overline{A_X}\big] \leq \binom{q}{2} 2^{-n}, \text{ and } \Pr\big[\overline{A_S}\big] \leq \binom{q}{2} 2^{-n}. \tag{A9}$$

Substituting Equation (A9) by Equation (A6),

$$1 - \Pr[A_X \cap A_S] \leq 2\binom{q}{2} 2^{-n} = \frac{q(q-1)}{2^n} < \frac{q^2}{2^n}. \tag{A10}$$

Call Equation (A5)

$$\left| \Pr\Big[ G\big(\eta_3(f)(L_1,R_1),\ \cdots,\eta_3(f)\big(L_q,R_q\big)\big) = 1 : f \overset{U}{\leftarrow} \mathcal{F} \Big] - P_G \right| \leq \frac{q^2}{2^n}.$$

□

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
