# Peer review of "New Mathblocks-Based Feistel-Like Ciphers for Creating Clone-Resistant FPGA Devices"

_cryptography, doi:10.3390/cryptography3040028_

Round 1
Reviewer 1 Report
Check English grammar (For example, line 29, electronic devices is identify: this sentence is wrong).
Very descriptive which is good.
Very nice figures though I would make my own without copying them.
The manuscript design is nice and clear I would need more information on the conclusion sections after all this work that has been done, not very clear and round.
Reviewer 2 Report
The paper investigate the digital PUF scheme, called as Secret Unknown Cipher. The idea is to develop unknow cipher functions for generating randomness. The structure of the scheme and FPGA implementation are clearly described. In addition, the mappping strategy for creating SUCs is also introduced. The overall idea is interesting and the description of FPGA implementation are reasonable.
However, the manuscript needs a substantial improvement. The reviewer’s
comments and suggestions are as follows:
General comments:
Minor comments:
Secret Unknown Ciphers SUCs should be changed to Secret Unknown Ciphers or SUCs. They are the same words. For exmaple, in page 2, line 59 and page 1, line 19. Fig. 19 should be explained with more details. What do the different markers and the bigger blue circles mean?Author Response
Please see the attachment
